# Physicochemical Characteristics and Dynamic Charge Mapping in Thermally Aged Two-Layered Polymer Considering Surface States: Experiment and Simulation

**DOI:** 10.3390/polym12030634

**Published:** 2020-03-10

**Authors:** Xiongwei Jiang, Wenxia Sima, George Chen, Qingjun Peng, Potao Sun

**Affiliations:** 1State Key Laboratory of Power Transmission Equipment & System Security and New Technology, Chongqing University, Chongqing 400044, China; cqsmwx@cqu.edu.cn (W.S.); sunpotao@cqu.edu.cn (P.S.); 2School of Electronics and Computer Science, University of Southampton, Southampton SO16 1BJ, UK; gc@ecs.soton.ac.uk; 3Electric Power Research Institute of Yunnan Power Grid Corporation, Kunming 650106, China; pengqingjun666@163.com

**Keywords:** multilayered, physicochemical, thermal aging, bipolar charge transport, sensitivity analysis, surface states

## Abstract

Under operational conditions of high electric fields and elevated temperatures, the accumulation of space charges at multilayer insulation interfaces is often considered as an important factor affecting insulation performance. This study experimentally explored the influence of different thermal aging degrees (110 °C for 0, 720, 1600, 2100, and 2900 h) on physicochemical characteristics. The space charge dynamics in two-layered thermally aged PET-PET films were measured using the pulsed electro-acoustic (PEA) method and simulated on the basis of a one-dimensional modified bipolar charge transport model. The parameterization for key parameters involved in the model was analyzed through parameter sensitivity. Results indicated that the molecular structure, crystallinity, and dielectric spectra of the PET films are affected by thermal aging. The thermalization process also has noticeable effect on the surface state characteristics, which are characterized by deeper trap depth and larger trap density. Several experimental phenomena measured by the PEA method were observed on the basis of numerical simulation.

## 1. Introduction

Multilayered polymeric insulation has been widely used to meet specific requirements in electrical equipment, such as dry-type reactors and gas-insulated transformers. Layered insulation hinders the breakdown process by providing localized states at layer interfaces to the propagation of the breakdown pathway. 

However, insulation performance degrades under combined thermal, electrical, and mechanical stresses during long-term operation. Under a high DC field, dielectric–dielectric and metal–polymer interfaces combine a potentially mechanically weak boundary with substantial interface (surface) charges and a corresponding large field gradient [1,2,3,4,5]. Therefore, electrical failure likely occurs at these particular positions. The way in which space charge trapping/detrapping occurs at the interface must be investigated. In terms of experimentation, the majority of studies investigated the effects of thermalization [1], electrical aging [5], voltage polarity reversal [4], temperature [3], and moisture [2] on the charge dynamics in multidielectrics. 

As for simulations, as early as 2005 [6], a dynamic model of space charge distribution in an XLPE-EPR system was proposed by S. Le Roy et al.; it features the injection of both electrons and holes from two electrodes and charge trapping in traps distributed exponentially in trap depth. The existence of surface states in polymeric materials was experimentally demonstrated by Mizutani in 2006 [7]. However, the extension of surface area is affected by many factors. In S. Le Roy’s works, the surface area is only 1 μm in depth, and a Gaussian filter is applied to the simulated data to produce profiles of the same form as the experimental ones [8,9]. In 2013, the origin and characteristics of surface state were briefly described by S. Le Roy et al. Surface state is defined as a local energy state (trap) on the surface of an insulator or at the interface between two materials (dielectric–dielectric and metal–polymer) [9]. In 2014 [10], K. Wu et al. simulated the interfacial charges between oil and paper and showed that the formation of interfacial charges can be attributed to the charge barrier (deep traps) existing at the interface, and that the deep traps may be related to surface condition, pressure, and moisture. In the same year, Meng H. Lean et al. described a hybrid algorithm for the solution of drift-diffusion equations for bipolar charge transport in layered polymer films, but only simulated charge profiles were present [11]. In 2016 [12], Y. Yin et al. preliminarily simulated the space charge behavior between two insulations of similar nature and introduced deeper electron traps or obstacles to the electronic transport via a bipolar charge transport (BCT) model.

Despite the efforts exerted to understand the space charge characteristics in layered dielectric systems, the majority of studies were conducted experimentally [13], simulation in similar two-layered polymers was rarely involved, and little attention was paid to the characterization of surface state and its influence on interfacial charges. The physical interfaces constituted by the association of two identical dielectrics do not comply with the usual Maxwell–Wagner–Sillars behavior and surface state must be considered. The surface state plays a critical role in charge injection, accumulation, and the drift-diffusion process [14]. In those available references, the surface states only affect unipolar charges for simplicity of modeling. However, the polarity of space charges at the interface is the result of superposition of bipolar charges [11]. It also brings great complexity to the simulation of the effect of thermal aging on surface states and charge characteristics during polarization and depolarization processes due to the weak universality of the existing model.

In the present work, we physicochemically characterized thermally-aged PET films and studied the effect of thermal aging on surface states and charge dynamics in PET–PET two-layered insulation during polarization (15 kV/mm) and depolarization processes. We proposed an improved model based on the classic BCT model to verify the accuracy of the analysis [15]. For quantitative analysis, parameterization for key parameters involved in the model was conducted via sensitivity analysis. We also simulated some observed experimental phenomena, particularly for fast injection of the charge, charge dissipation, and interfacial charge trapping/detrapping processes. 

## 2. Experimental Procedure

### 2.1. Sample Preparation

PET particles were firstly heated to molten state and poured into a metal mold with a thickness of 200 ± 5 μm. To prepare films, then the mold was thermally pressed at 393 K with a pressure of 15 MPa using a press vulcanizer. Lastly the single-layer PET film was made. The thickness of the film was measured by a spiral micrometer. Subsequently, films were thermally aged at 110 °C in a vacuum chamber (<10 Pa) for 0, 720, 1600, 2100, and 2900 h. There are two layers of aluminum clapboards in the vacuum chamber. During the thermal aging process, the polymer films were placed on aluminum clapboards.

### 2.2. Physicochemical Characterization

Fourier transform infrared spectroscopy (FTIR, Nicolet 6700, Thermo Electron Scientific Instruments Corp. Franklin, MI, US), X-ray diffraction (XRD, Bruker company, Beijing, China), and dielectric spectra measurements (Novocontrol concept 40, Novocontrol GmbH, North Wales, Germany) were performed at 293 K to examine the influence of thermal aging on dielectric physicochemical characteristics. 

#### 2.2.1. Molecular Structure

The effect of thermal aging on the molecular structures of the PET films was analyzed by FTIR, which was operated on a reflective mode for the results to only reflect the changes in molecular chain at a certain depth on the surface. The spectra between 400 and 4000 cm^−1^ were measured. The background of the atmosphere was measured and subtracted from each spectrum.

#### 2.2.2. Crystallinity

We performed XRD at a scan rate of 4° per min to observe the crystallization of thermally aged PET films over angles ranging from 5° to 50°. The filter plate was made of Ni. The applied voltage was 40 kV, and the current was set at 40 mA. 

#### 2.2.3. Dielectric Spectra

Dielectric relaxation spectroscopy of thermally aged PET films was performed using Novocontrol broadband dielectric spectra. The measured frequency ranged from 0.01 to 10^6^ Hz.

### 2.3. Space Charge Measurement

Before the pulsed electro-acoustic (PEA) experiment, the samples were placed in a vacuum chamber (323 K) and short-circuited for 24 h, which is beneficial for removing moisture and residual charges in the films. Two single-layer PET films were mechanically contacted by mechanical force between the upper and lower electrodes. The principle of PEA measurement is described as follows [16]: the pulsed voltage, together with the poling DC voltage, was applied across the two-layered PET films. The application of the pulse modified the electrostatic force distribution acting on the sample and generated acoustic waves. These waves were detected by a piezoelectric sensor (PVDF) after having traveled through the sample and grounded electrode. The sensor converted the signals into electrical signals. The amplitude of the converted signals was proportional to the charge density, and the space-time distribution characteristics of the charges was inferred from the signal propagation time. All PEA experiments were conducted under a DC field of 15 kV/mm and at 333 K.

## 3. Results and Discussion

### 3.1. Physicochemical Characterization

Figure 1 shows the FTIR spectra of thermally aged PET films. With the increase in thermal aging time, the reflection intensities at 1018 and 1097 cm^−1^ wavenumbers weakened, which corresponded to the symmetrical aromatic C–O and asymmetrical aliphatic C–O stretching, respectively [17]. The characteristic peak at 1713 cm^−1^ (C=O) was the most important concern in the spectra. The quantity of C=O bonds slightly increased as thermal aging continued, indicating that the molecular chains on the PET surface area were broken and oxidized under thermal aging.

The XRD results in Figure 2a demonstrated that the intensity of diffraction peak varies with thermal aging time. Three characteristic diffraction peaks located at 2θ equaling to 12, 22.5, and 26 degrees were found. The diffraction peak presented a flat shape, revealing the nature of amorphous region, and the sharp characteristic diffraction peak is attributed to the crystal region [18]. This result indicated that the PET is a semicrystalline film. The crystallinity *X_c_* can be calculated based on Equation (1), and the results are shown in Figure 2b.
(1)$$X_{c}=1-\frac{S_{a}}{S_{a}+S_{c}}\times 100\%$$
where *S_c_* is the sum of crystallization peaks area, and *S_a_* is the amorphous peak area. 

The crystallinity *X_c_* initially decreased and then increased with the increase in thermal aging time, suggesting that at the initial stage of aging, the thermal cracking of molecules destroyed the crystallization area to some extent. As thermal aging continued, affecting the dielectric bulk further became difficult. As a result, the molecular thermal cracking reached a steady state, as shown in Figure 1. The longer chains in homopolymers may generate greater restraints to form large crystals; when some parts of the molecular chain break, the recrystallization process begins and *X_c_* increases [19].

Figure 3 shows the dielectric relaxation spectra of thermally aged PET films. The relative permittivity *εr* in Figure 3a initially increased and then decreased with the increase in aging time. *εr* reflects the density of polarizable units (space charge, dipoles, etc.) in the dielectric, and its variation indicated that the polarizable components increased during the initial thermal aging. However, during the later stage of aging, free radicals were oxidized and gasified, as represented by the decrease in C=O bond numbers illustrated in Figure 1. In addition, the increase in crystallinity inhibited dipole polarization to some extent. Therefore, *εr* initially increased and then decreased with aging time. It also exhibited a decreasing trend as the measurement frequency increased. The reason is that all free dipoles and polarizable groups in the chains can orient themselves at low frequencies, leading to increased permittivity value and weak orientational polarization as the frequency increased due to longer orientation time [20].

The effect of thermal aging on dielectric loss tanδ is shown in Figure 3b. As the relaxation time of space–charge polarization varies in the range of 0.1–100 s [20] and dipolar relaxation usually occurs approximately between 1 and 100 MHz depending on the nature of dipoles, 10 Hz is approximately speculated to be the dividing line of interfacial polarization (space–charge polarization) and dipolar relaxation. Considering the test errors, the thermal aging had minimal effect on tanδ in the high-frequency region (Table 1). In the low-frequency range, tanδ initially decreased and then increased as thermal aging time increased; this result is consistent with the variation in dielectric crystallinity under thermal aging. The interface between crystal and amorphous region can provide large numbers of trap centers. Thus, space charge polarization loss varied with dielectric crystallinity and affected tanδ in the low-frequency region.

### 3.2. PEA Measurement Results

Figure 4 shows the evolution of space charge behavior during polarization and depolarization processes. As demonstrated in Figure 4a, a large amount of charges was formed in the bulk (trapped or mobile) once the voltage was applied. Fast charges were also measured in Thomas et al.’s work, wherein a fast front of positive charges was observed in LDPE shortly after applying voltage [21]. In the present work, the density of negative charges at the PET–PET interface increased with polarization time. When the depolarization process was started (two electrodes were short circuited and grounded), the interfacial charges were difficult to dissipate due to the effect of surface state. The electrons in the bulk near the cathode side dissipated rapidly, leading to a certain amount of holes. This phenomenon is probably due to the difference in dissipation rates of the electrons and holes. In addition, some electrons decayed slowly in the second layer, suggesting that the trap depth in the bulk is multiple. 

Similar charge dynamics were observed with the increase in thermal aging time, as shown in Figure 4c. After polarization for 1800 s, the interfacial charge density increased from −5.30 C/m^3^ to 5.42 C/m^3^ compared with Figure 4a. In addition, the accumulation of electrons in the two layers was more asymmetrical. In particular, more electrons accumulated in the layer near the anode (the average charge densities in the left and right layers were −0.77 and −1.68 C/m^3^, respectively). This phenomenon was also observed in a study of two-layered XLPE films by F. Rogti et al [3]. When the injected charges transferred to the second layer, they lost some kinetic energy upon crossing the interface due to the existence of surface state. The carrier mobility decreased and the charges were easily trapped by deeper traps. As a result, the charges in the second layer dissipated slowly during depolarization. As thermal aging deepened, the interfacial charge density increased from −5.42 C/m^3^ to 6.20 C/m^3^, as displayed in Figure 4e. This finding indicated that the trap density in the surface area increased, consistent with the research of G. Chen who found that at the early stage of thermal aging in the air, a small number of deep traps were introduced near the surface of LDPE films [22].

As shown by the charge profiles in Figure 4a,c,e during polarization, the charge density at the cathode initially increased and then decreased with the increase in polarization time. This trend can be attributed to electron injection and electrons moving into the bulk as polarization time increases [23]. 

In addition, the distribution shape of the interfacial charges gradually changed from inverted “U” to positive “U” shape as the polarization continued, which demonstrated that a physical interface can act as a trapping zone for both types of carriers and that this interface is a preferential site for trapping, provided that the flux of charges is high enough [3]. 

The location of the interfacial charge peak was not strictly at the interface (located at 200 μm); it gradually moved towards the anode (from the position of 190 to 208 μm) as the polarization continued. The influence of surface states (on both sides) on the processes of charge leaving or entering from one dielectric to another is likely different, and the capture ability of the interface varies with charge flux [3]. Therefore, due to the increased electron penetration in the bulk up to the interface as polarization continued, these electrons partly compensated or recombined the initial positive charges, and the superposition of positive and negative charges in this area led to the shift in peak position. 

When summarizing all charge profiles during depolarization in Figure 4b,d,f, we mainly focused on the evolution of charge behavior at the PET–PET interface. Intriguingly, the negative charges at this interface initially increased and then gradually decreased with depolarization time. The difference in interfacial charge density during initial depolarization and polarization for 1800 s was not evident, which was different from the decay phenomenon in the bulk. The characteristics of the surface states, that is, deeper trap depth and larger trap density, are experimentally and theoretically supported by relevant literature [10,12,14]. This phenomenon was simulated in Section 4.4.

## 4. Numerical Analysis of Charge Dynamics

To verify the above analysis of charge dynamics in two-layered PET films, we based the modeling of BCT in layered PET films on the well-known one-dimensional BCT model [15]. The following features were added to the model: the injected charges being trapped in traps were exponentially distributed in the energy as a function of space near the surface area (described in Section 4.1) [6,9]; both fast and slow electronic carriers (electrons and holes) were injected into the polymer from the electrodes; the bipolar drift-diffusion equations for charge transport were time-space dependent; and no charge extraction barrier existed at each electrode, as described in Section 4.2. 

The parameter sensitivity analysis and setting and simulation results are presented in Section 4.3 and Section 4.4, respectively. For the sake of simplification, only the injected charges were considered, and capacitive and image charges were not considered in our initial model. 

### 4.1. Surface States

Based on a previous model [9,15], an interface region at each dielectric–dielectric and metal–polymer interfaces was introduced in the improved model, accounting for localized states due to physical and chemical disorders in these regions. The extension in the surface region is reported as being orders of μm in dielectrics [24]. In the context of polymers, surface states may be influenced by many factors, such as the presence of impurities, diffusion of by-products, or imperfection of the surface (polishing or surface contamination and oxidation). Therefore, a deeper region was involved in the modeling when the surface states were related to residues or oxidation, such as in the case of thermally aged PET films in the present work. A schematic of the energy state distribution and grid division used for simulation is provided in Figure 5. 

Surface states were present but their complete description in terms of density, extension in space, and energy distribution remained and was excessively complicated. The surface region was characterized by a higher density of trapping states and an exponential distribution of deeper trap levels as a function of space. The corresponding detrapping coefficient in the interfacial region is expressed by Equation (2) as follows:(2)$$D_{t}{}_{\left(e,h\right)}=\nu_{ATE}\cdot \exp(-E_{T(e,h)}/kT)/\exp{\left(\left|x_{i+m}-x_{i}\right|\right)}^{n}$$
where *D_t_* is the detrapping coefficient for trapped electrons or holes; subscripts *e* and *h* refer to electrons and holes, respectively; *v_ATE_* is the attempted escape frequency for trapped charges; *E_T_* is the trap depth at the edge of surface state; *k* is the Boltzmann’s constant; *T* is the temperature; *x_i_* is the location of the interface; *d_s_* = |*x_i_*_+*m*_ − *x_i_*| is the extension in the surface region; and *n* is defined as asymmetry index. 

On the basis of two identical PET films submitted to the same thermal treatment, the surface states for one kind of carrier on both sides of the interface were similar. However, when charges crossed the interface, the injection barrier was different from the extraction barrier, which was caused by the microcavities at the interface between two mechanically contacted PET films or electrode/PET film. An extraction or injection barrier is added only in one calculation grid cell at the very surface, in order to simulate the process of charges crossing the interface; the simulation leads to an accumulation of charges (trapped or mobile) only in this grid cell. The injection and extraction barriers, with exponential distribution as a function of space, were added to the surface region (not limited to one calculation grid cell) to eliminate this numerical problem. Therefore, the surface state distribution considered both the local energy states at the surface and the difference between the injection and extraction barriers. Thus, the distribution of trap depth on the left and right sides of the interface was asymmetrical. This distribution was achieved by adjusting *n* (*n*_1_ for charges leaving the surface and *n*_2_ for charges entering the surface), which was necessary as the interfacial charge distribution was asymmetrical, as shown in Figure 4 and Reference [3]. For each kind of species, the interfacial region was present at each electrode and PET–PET interface. This region affected the penetration of charges into the bulk, accumulation at the PET–PET interface, transporting to the second layer, and the extraction process from the electrode. The bulk region is defined as a single trap level *E_T_* with a constant trap density for electrons and holes. Although the PEA results showed deeper traps in the bulk, these traps were difficult to realize in the modeling process. Therefore, we used a balanced value in the trap depth for simplicity.

### 4.2. Charge Generation and Transport 

The carrier injection was assumed to follow the Schottky mechanism [25]:(3)$$j_{e}(0,t)=r_{e}AT^{2}\exp(-\frac{W_{ei}}{kT})\exp(\frac{e}{kT}\sqrt{\frac{eE(0,t)}{4\pi \varepsilon_{0}\varepsilon_{r}}})$$
(4)$$j_{h}(d,t)=r_{h}AT^{2}\exp(-\frac{W_{hi}}{kT})\exp(\frac{e}{kT}\sqrt{\frac{eE(d,t)}{4\pi \varepsilon_{0}\varepsilon_{r}}})$$
where *j*_e_(0, *t*) and *j_h_*(*d*, *t*) are the injected current densities for electrons at the cathode (*x* = 0) and holes at anode (*x* = *d*), *A* is the Richardson constant 1.2 × 10^6^ Am^−2^K^−2^, *W_ei_* and *W_hi_* are the effective injection barriers for electrons and holes, and *E* is the electric field at the electrode. During PEA measurement, quantities of charges reached the opposite electrode within a few seconds after polarization. Therefore, two types of carriers (both fast and slow mobility for electron and hole) were considered in the simulation. *r_e_* and *r_h_* represent the portion of injected charges with large mobility for electrons and holes, respectively. Their values were between 0 and 1. 

The bipolar transport considered herein is drift-diffusion, described as a conduction process governed by constant mobility and diffusion. The injected bipolar charges drifted and diffused through the layered films and then were trapped/detrapped and recombined under the action of electric fields. The time-space dependent equations describing the charge dynamics are as follows [26,27]:(5)Drift-diffusion: j(x,t)=μE(x,t)n(x,t)−D∇n(x,t)
(6)$$\mathrm{Poisson}:\;\frac{\partial E(x,t)}{\partial x}=\frac{\rho (x,t)}{\varepsilon_{0}\varepsilon_{r}}$$
(7)$$\mathrm{Convection-Reaction}:\;\frac{\partial n(x,t)}{\partial t}+\frac{\partial j(x,t)}{\partial x}=s_{i}(x,t)$$
where *j*(*x*, *t*) is the total convection flux; *n*(*x*, *t*) is the carrier density, representing electrons and holes; *D* is the diffusion coefficient; *ρ* is the net charge density, including mobile and trapped electrons and holes; and *s_i_*(*x*, *t*) is the source term representing the charge trapping/detrapping process and recombination between positive and negative charges [26,27], which can be introduced as follows:(8)$$s_{1}=\frac{\partial n_{e\mu }}{\partial t}=-B_{tr(e)}n_{e\mu }(1-\frac{n_{et}}{N_{Te}})+D_{te}n_{et}-S_{1}n_{ht}n_{e\mu }-S_{3}n_{hu}n_{e\mu }$$
(9)$$s_{2}=\frac{\partial n_{et}}{\partial t}=B_{tr(e)}n_{e\mu }(1-\frac{n_{et}}{N_{Te}})-D_{te}n_{et}-S_{2}n_{h\mu }n_{et}-S_{0}n_{ht}n_{et}$$
(10)$$s_{3}=\frac{\partial n_{h\mu }}{\partial t}=-B_{tr(h)}n_{h\mu }(1-\frac{n_{ht}}{N_{Th}})+D_{th}n_{ht}-S_{2}n_{h\mu }n_{et}-S_{3}n_{h\mu }n_{e\mu }$$
(11)$$s_{4}=\frac{\partial n_{ht}}{\partial t}=B_{tr(h)}n_{h\mu }(1-\frac{n_{ht}}{N_{Th}})-D_{th}n_{ht}-S_{1}n_{ht}n_{e\mu }-S_{0}n_{ht}n_{et}$$
where *n_e_* and *n_h_* include both fast and slow carriers; *S_i_* is the recombination coefficients; *B_tr_*_(e)_ and *B_tr_*_(*h*)_ are the trapping coefficients for electrons and holes, respectively; and *N_T_* is the trap density. 

Parameters such as *B_tr_* and *N_T_* were rewritten as *F*_(*e, h*)_*B_tr_* and *F*_(*e, h*)_*N_T_*, respectively, to represent the larger trap density in the interface regions between PET–metal and PET–PET. The optimization coefficient *F*_1_ represented the situation of charge leaving the surface and *F*_2_ denoted the charges entering the surface. *F*_2_ was slightly larger than *F*_1_, and their values were influenced by thermal aging. In the surface region, *D_t_* was modified with an exponential function (deeper trap level), as shown in Equation (2). 

As for charge mobility, the high mobility for fast electrons and holes can be extracted from the experimental results (Figure 4) of our first attempt. Under the assumption of a uniform field and considering the travelling time of the front carriers and the distance between the electrodes, we can obtain the mobility values of the order of 9 × 10^−12^ m^2^V^−1^s^−1^ for both fast electrons and holes. The mobility of charge carriers is related to the electric field, as shown in Equation (12) [28].
(12)$$\mu_{\left(e,h\right)}=\mu_{0}E^{c-1}\;(c>1)$$
where *μ*_0_ is a constant and *c* is a fixed component (*c* = 1.17). The influence of electric field on mobility was limited in our modeling; thus, the mobility was set to be a constant. The small mobility for the rest of the electrons and holes was set as 2 × 10^−16^ and 1 × 10^−16^ m^2^V^−1^s^−1^, respectively, which is reasonable for insulating polymers [29]. 

As for charge extraction, S. Le Roy’s work [25] and the present paper confirmed that when an extraction coefficient or extraction barrier is added, the simulation leads to an accumulation of charges (trapped or mobile) only in the last cell of the calculation grid. To solve this numerical problem, we only considered the effects of surface state and diffusion on extraction process. Therefore, the extraction current densities for electrons and holes followed the drift-diffusion equations as follows:(13)$$j_{e}(d,t)=\mu_{e}E(d,t)n_{e\mu }(d,t)-D\nabla n_{e}(d,t)$$
(14)$$j_{h}(d,t)=\mu_{h}E(0,t)n_{h\mu }(0,t)-D\nabla n_{h}(0,t)$$

### 4.3. Parameter Sensitivity Analysis and Setting

Parameter sensitivity analysis was performed to quantify the parameter values and meet the modeling requirements. The influence of traditional parameters in models of single layer dielectric, such as injection barrier and trap density, was not explored. Instead, we paid attention to the characterization parameters of surface state. The effects of bulk trap depth *E_T_* and fast mobility *m_uf_* on charge characteristics were also discussed. All simulation results after polarization for 1800 s were also presented. The condition of thermal aging for 720 h was used as the basis of modeling. All other parameters were kept constant if not specifically indicated, except the parameter under sensitivity analysis, to clearly observe the influence of a specific parameter during simulation. 

#### 4.3.1. Influence of Surface State Characterization Parameters

As described in Section 4.1 and Section 4.2, surface states are mainly characterized by the modulated interfacial trapping coefficient *F*_(*e, h*)_, asymmetry index *n*, and extension of surface state *d_s_*. Their effects on the accumulated charges are shown in Figure 6a–c. The legends marked with blue dots are the adopted parameter values. 

As shown in Figure 6a, when *F*_(*e*)_ increased, the dominant charge polarity at the interface gradually evolved from bipolar into negative, and the charge density at the interface increased. However, the charge density in the bulk slightly decreased. Thus, the accumulation of negative charges enhanced the electric field at the anode (increased the hole injection flux) and weakened the electric field at the cathode (reduced the electron injection flux), resulting in reduced net charge density in the bulk. 

The influence of index *n* on the simulation results is presented in Figure 6b. Equation (2) demonstrates that when the depth of surface state *d_s_* remains unchanged, the increase in *n* means the increase in surface trap depth. Intriguingly, when *n*_1_ and *n*_2_ increased simultaneously at a rate of 20%, the charge density at both electrodes and the PET–PET interface gradually increased and stabilized, which was different from the influence of *F***_(*e*)_** on interfacial charge dynamics. The influence of increasing interfacial trap depth was limited when *F***_(*e*)_** was constant. In addition, as *n* increased, the position corresponding to the peak density gradually moved towards the anode. 

Figure 6c shows the influence of the extension of surface state *d_s_* on charge dynamics. As *d_s_* changed, *n* also changed correspondingly, making the range of trap depth in the surface region consistent. As *d_s_* increased, the charge distribution at the interface (electrode/PET and PET/PET) became wider, the interfacial charge density gradually increased, and the position of charge peak moved towards each electrode.

#### 4.3.2. Other Factors Affecting Charge Characteristics

The influence of *m_uf_* for fast bipolar charges on the simulation results is shown in Figure 7a. The increase in *m_uf_* led to a remarkable reduction in net charge density (negative) in the bulk and at the PET–PET interface. The charges with slow mobility are believed to be more easily trapped. Figure 7b shows the influence of *E_T_* on charge dynamics. The charge density in the bulk increased significantly as *E_T_* increased. However, the charge density at the PET–PET interface began to decrease slightly when *E_T_* increased to a certain extent. When comparing Figure 7 with Figure 6, both *m_uf_* and *E_T_* showed more evident influence on the charge distribution in the bulk than that of the characterization parameters for surface state. The charges with smaller mobility travelling in the bulk with deeper trap depth are more beneficial to the charge trapping process. Thus, the experimental phenomenon in Figure 4, that is, charges are asymmetrically accumulated in two layers, can be attributed to the reduced mobility when charges across the interface (a series of trapping and detrapping) are being trapped by deeper traps in the second layer.

To sum up, the appropriate values for *n*_1_, *n*_2_ and *d_s_* can be determined according to the peak position and distribution area at the interface. *F* should be adjusted according to the interfacial charge density under thermal aging, although *m_uf_* also affects the interface charge density, with *F* as an essential factor affecting the effective mobility of charges. Both *m_uf_* and *E_T_* should be set according to the charge distribution characteristics in the bulk. The important parameters used for modeling after a series of parameters sensitivity analysis are listed in Table 2.

### 4.4. Results of BCT Model

#### 4.4.1. Simulation Results under 15 kV/mm 

Figure 8 shows the simulated charge profiles as a function of space and time during polarization and depolarization process for 1800 s, respectively. 

During the polarization process, the charge density at both electrodes showed a gradually increasing trend with the charge injection process, as no capacitive and image charges were considered in the simulation. The variation in charge density at the two-layered PET interface greatly improved when diffusion process was considered during polarization. The phenomenon that interfacial peak position extends to another layer was observed by asymmetric parameter settings, such as trap energy and density for electrons and holes. When the electrons and holes passed through the interface via drift and diffusion, they could be easily captured and had more difficulty escaping on the second layer surface due to the asymmetrical surface state (*F*_2_ > *F*_1_, *n*_2_ > *n*_1_). The superposition of bipolar charges made this phenomenon more evident. With the increase in thermal aging time, the increased interfacial negative charge density can be attributed to the increased deep trap density of the electrons at the PET–PET interface. The asymmetrical distribution of electrons in the two layers was observed through the following two main steps: first, we set a lower injection barrier for fast electrons, which means fast electrons are more easily injected from the cathode than fast holes and play a leading role in the bulk. Second, we reduced the mobility of electrons and holes after they crossed the interface because some kinetic energy was lost.

A small quantity of holes distributed in the layer close to the cathode may be caused by the injected holes being trapped during polarization or detrapping and drifting of interfacial holes during depolarization. When the dissipation rates of holes and electrons at the interface were different, the superposition effect of bipolar charges led to the net negative interfacial charge density increasing first and then gradually decreasing with the increase in depolarization time. As for the charge decay phenomenon in the bulk, although only a single trap level *E_T_* was considered in this area, and it is usually treated similarly in most literature [4,6,15], not all charges in the bulk were trapped charges. Some free charges in the layer near anode dissipate quickly and then remain stable, which is very similar to the present experiment. 

#### 4.4.2. Simulation Results under 10 kV/mm 

The model was also used to simulate the charge dynamics at 10 kV/mm to further verify its reliability. The PET films thermally aged for 720 and 2900 h were studied, and the simulation results are shown in Figure 9a,c, respectively. Figure 9b,d show the corresponding PEA results. When the applied field was reduced to 10 kV/mm, the interfacial charge densities in Figure 9a,c decreased after polarization for 1800 s compared with those in Figure 8a,e. The corresponding test results showed similar accumulation characteristics at each polarization moment, and the interfacial charge density increased after aging for 2900 h.

The charge transport mechanism in a wide variety of disordered and amorphous materials is multiple trapping through a series of jumps between localized states or shallow traps. The charges in deep traps may escape when carriers gain enough energy, for example, through phonon interaction in thermally assisted hopping, to overcome a potential barrier of localized state and enter an extended state. They may also move via phonon-assisted tunneling through a potential barrier between deep traps [31]. Therefore, the mechanism of charges detrapping from deep traps is complicated, which entails more difficulties to the simulation during depolarization process. To verify the accuracy of our analysis for the test in Section 3.2, we considered the asymmetrical surface states and fast charge dynamics in the simulation and modified several parameters that may be related to the thermal aging process. The reliability of the proposed model was verified by both polarization and depolarization processes for the first time.

## 5. Conclusions

In this study, FTIR, XRD, and dielectric spectra measurements were performed to gain valuable insights into the effect of thermal aging on dielectric physicochemical characteristics. The charge dynamics during polarization and depolarization processes were measured using the PEA method and simulated on the basis of the modified BCT model to verify the accuracy of the analysis. The following conclusions are obtained:

The molecular chains on the PET surface area are broken and oxidized (C=O increases) under thermal aging. With the increase in thermal aging time, crystallinity *X_c_* initially decreases and then increases due to the recrystallization process. *ε_r_* increases due to the increase in the polarizable unit density. In the low-frequency region, tanδ is also affected by the space charge polarization.

The asymmetrical parameter settings for fast electrons and holes, such as injection flux and surface states, were considered in the improved model. Parameter sensitivity analysis was performed for parameterization of the key parameters involved in the model.

The simulation results indicated that the surface state features with high trap density (*F*_1_, *F*_2_) and deep trap depth (*n*_1_, *n*_2_), exponentially distributed in the depth of surface state *d_s_*, are crucial to the accumulation of charges at the metal–PET and PET–PET interfaces. The decrease in *m_uf_* and increase in *E_T_* resulted in more charges being trapped in the bulk.

At present, the parameterization for the interfacial region for quantitative analysis remains a challenge because parameters, such as *F*_(*e*,*h*)_, *n*_(*e*, *h*)_, *d_s_*_(*e*, *h*)_, *m_uf_*, and *E_T_*, affect the interfacial charge characteristics. However, our research can still be applied to qualitative analysis even in the most complex case.

## Figures and Tables

**Figure 1 polymers-12-00634-f001:**
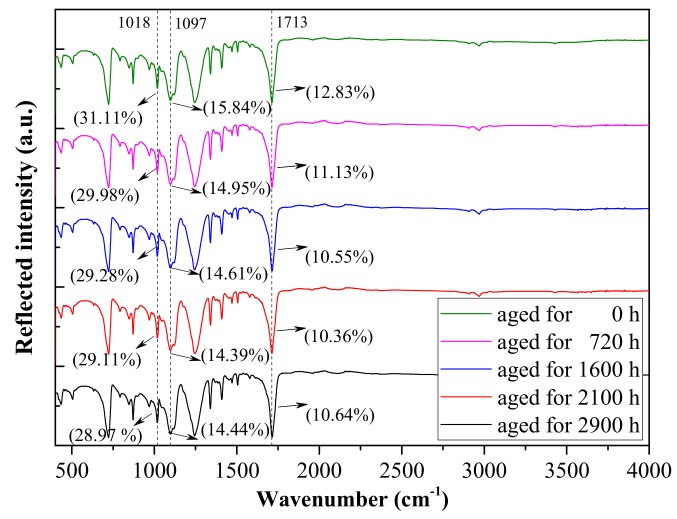
Fourier transform infrared (FTIR) spectra of thermally aged PET films aged at 110 °C.

**Figure 2 polymers-12-00634-f002:**
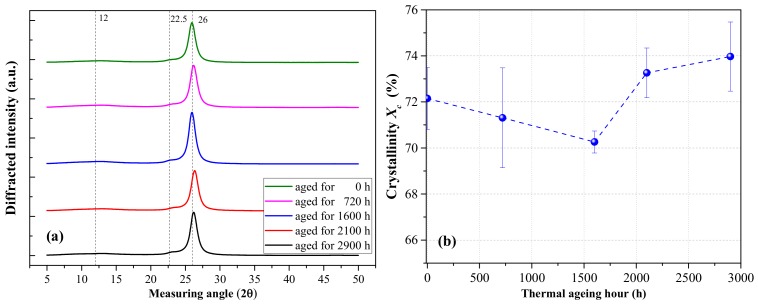
(**a**) X-ray diffraction (XRD) curves and (**b**) calculated crystallinity *X_c_* of thermally aged PET films aged at 110 °C.

**Figure 3 polymers-12-00634-f003:**
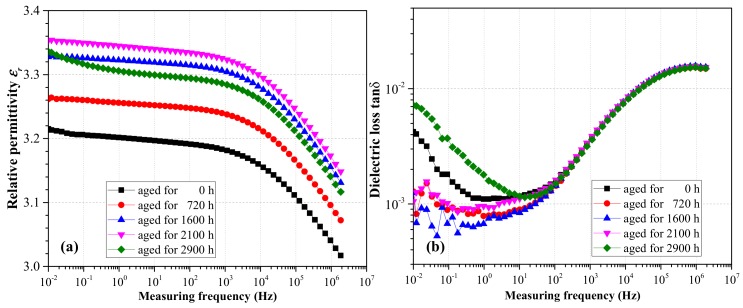
(**a**) Relative permittivity *ε_r_* and (**b**) dielectric loss tanδ of thermally aged PET films aged at 110 °C.

**Figure 4 polymers-12-00634-f004:**
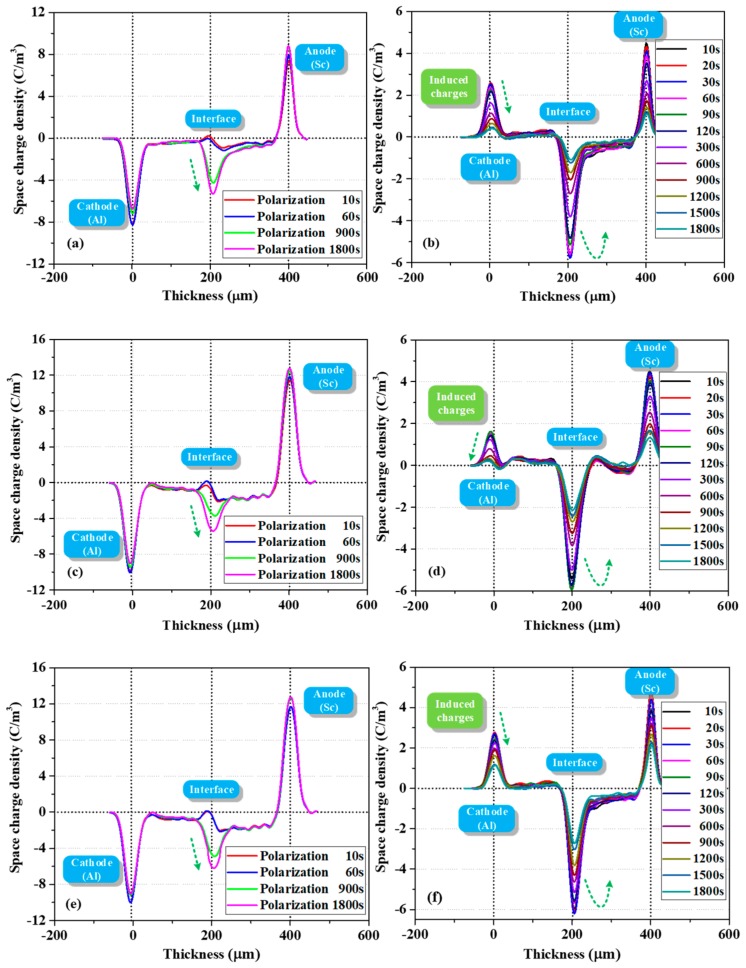
Evolution of space charge during polarization (left) and depolarization (right) processes of PET–PET samples with different thermal aging degrees (measured by PEA). Samples thermally aged for (**a**,**b**) 720, (**c**,**d**) 2100, and (**e**,**f**) 2900 h.

**Figure 5 polymers-12-00634-f005:**
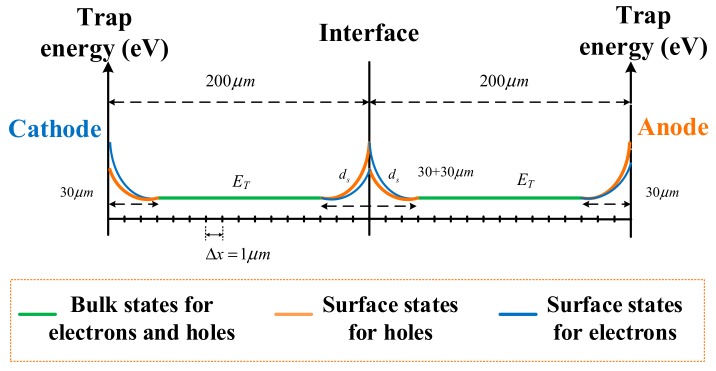
Schematic of the spatial grid and trap energy distribution used for simulations.

**Figure 6 polymers-12-00634-f006:**
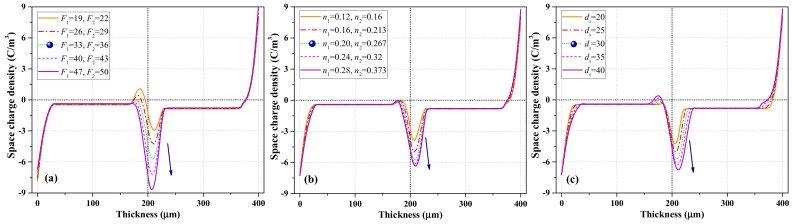
Simulation results of space charge after polarization for 1800 s of PET–PET samples. Influence of (**a**) *F***_(*e*)_** for electrons, (**b**) *n_(e, h)_*, and (**c**) *d_s_*_(*e*, *h*)_.

**Figure 7 polymers-12-00634-f007:**
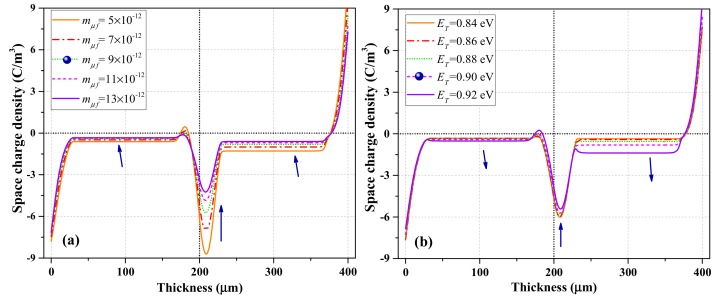
Simulation results of charge after polarization for 1800 s of PET–PET samples. Influence of (**a**) fast mobility *m_uf_* and (**b**) bulk trap depth *E_T_* for electrons and holes.

**Figure 8 polymers-12-00634-f008:**
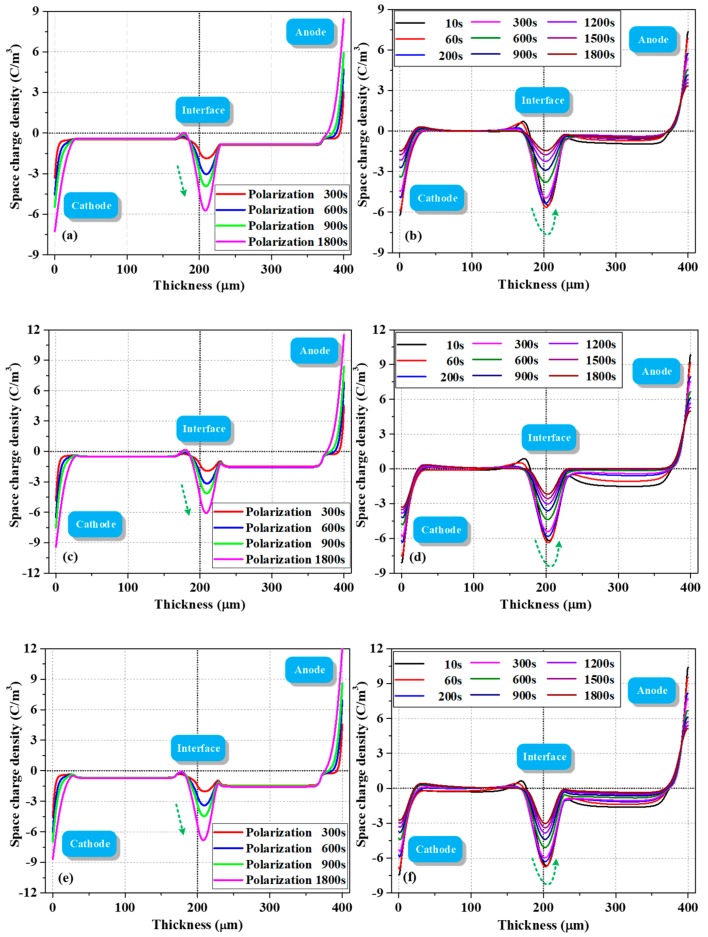
Simulation results of charge during polarization (left) and depolarization (right) processes of PET–PET samples, which were thermally aged for (**a**,**b**) 720, (**c**,**d**) 2100, and (**e**,**f**) 2900 hrs.

**Figure 9 polymers-12-00634-f009:**
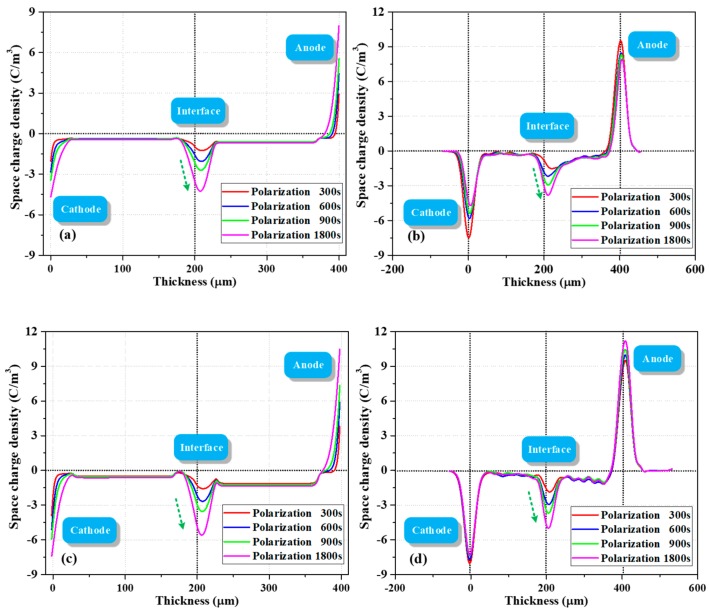
Comparison between simulation and test results of charge dynamics in PET–PET samples under 10 kV/mm. (**a**) Simulated and (**b**) corresponding results for 720 h, and (**c**) simulated and (**d**) corresponding results for 2900 h.

**Table 1 polymers-12-00634-t001:** Effect of thermal aging on dielectric loss tanδ in the low- and high-frequency regions.

Frequency		0.1 Hz	1 Hz	10^3^ Hz	10^5^ Hz
Dielectric loss tanδ	0 h	0.00181	0.00110	0.0037	0.0136
	720 h	0.00089	0.00079	0.0036	0.0134
	1600 h	0.00068	0.00067	0.0037	0.0138
	2100 h	0.00101	0.00096	0.0039	0.0138
	2900 h	0.00371	0.00180	0.0036	0.0133

**Table 2 polymers-12-00634-t002:** Definition of the modeling parameters. When the influence of thermal aging is considered, several parameters are provided within a certain range.

Parameters	Descriptions	Values
*W_sei_*	Injection barrier for slow electrons (eV), influenced by thermal aging	1.125–1.14 [28]
*W_shi_*	Injection barrier for slow holes (eV), influenced by thermal aging	1.155–1.16 [28]
*W_fei_*	Injection barrier for fast electrons (eV)	0.96
*W_fhi_*	Injection barrier for fast holes (eV)	0.98
*E_T_* _(*e*)_	Bulk trap level for electrons (eV)	0.9 [30]
*E_Td_* _(*h*)_	Bulk trap level for holes (eV)	0.9
*N_te_*	Trap density for electrons (C/m^3^)	100 [28]
*N_th_*	Trap density for holes (C/m^3^)	100
*m_use_*	Slow electron mobility (m^2^V^−1^s^−1^)	2 × 10^−16^
*m_ush_*	Slow hole mobility (m^2^V^−1^s^−1^)	1 × 10^−16^
*m_ufe_*	Mobility for fast electron (m^2^V^−1^s^−1^)	9 × 10^−12^
*m_ufh_*	Mobility for fast hole (m^2^V^−1^s^−1^)	9 × 10^−12^
*D_s_*	Diffusion coefficients for slow bipolar charges, including free and trapped charges (m^2^/s)	2 × 10^−14^
*D_ff_*	Diffusion coefficients for free fast bipolar charges (m^2^/s)	1 × 10^−9^
*D_pft_*	Diffusion coefficients for trapped fast bipolar charges during polarization (m^2^/s)	1 × 10^−13^
*D_dft_*	Diffusion coefficients for trapped fast bipolar charges during depolarization (m^2^/s), influenced by thermal aging	5 × 10^−13^–11 × 10^−13^
*B_tr_* _(e)_	Basic trapping coefficients for electrons (s^−1^)	2 × 10^−4^
*B_tr_* _(h)_	Basic trapping coefficients for holes (s^−1^)	2 × 10^−4^
*S_i_*	Recombination coefficients (m^3^ C^−1^s^−1^)	0.004
*d_s_*	extension of surface state (μm)	30
*n*	asymmetry index (*n*_1_, *n*_2_)	(0.2, 0.2667)
*F* _(*e*)_	Modification of trap density (*F*_1_, *F*_2_) for electrons at PET-PET interface, influenced by thermal aging	(33, 36) (35, 40) (40, 43)
*F* _(*h*)_	Modification of trap density (*F*_1_, *F*_2_) for holes at PET-PET interface	(10, 40)
*r* _(*e*)_	Portion of injected fast electrons	0.8
*r* _(*h*)_	Portion of injected fast holes	0.8

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
