# Peer review of "Physicochemical Characteristics and Dynamic Charge Mapping in Thermally Aged Two-Layered Polymer Considering Surface States: Experiment and Simulation"

_polymers, 2020, doi:10.3390/polym12030634_

Round 1

Reviewer 1 Report

Although the authors have modified their paper following some of my recommandations, especially in giving extra data on physico-characterization of their thermally-aged samples, I am still wondering on which basis they have adopted the energy distribution of electrons and holes as a function of depth as shown in Figure 5. Again, on the basis of two identical PET films submitted to the same thermal treatment, one can expect the same energy-depth relationship for carriers in both films. This is not what is shown on their (new) Fig. 5 where as example the max in trap energy at the PET-PET interface is identical for electron in a layer and hole in the other layer. It seems to me rather unphysical.

I still think the paper cannot be published.

Reviewer 2 Report

The manuscript Physico-chemical Characteristics and Dynamic Charge Mapping in Thermally Aged Two-layered Polymer Considering Surface States: Experiment and Simulation by Xiongwei Jiang and co-workers discusses the characteristics of charge transfer on aged two-layered polymer interfases. The work is original, reasonable well presented and of interest to the wider community of people who work with polymer insulating materials. I believe that overall the work has merit to be published and I will support this view.

Although the work presented is good I have a quple of minor points that can be considered by the authors should they wish.

In Figure 3 the loss tangent (tanδ) apears not to follow a trend as the overall aeging time increases. I fully understand that this is an experimental measurement and therefore difficult to eliminate all errors but I think the authors could expand a bit on this topic. If a trend is to be expected it would be nice to have a few lines where they discuss its absence.  On the same graph the authors discuss about the effects of DC conductivity but there is no indication on how it changes as a function of aeging time and frequency. I think a graph would be beneficial to the reader as it will provide far more conceptual understanding in comparison to the relative permitivity that is shown. Before Equation 13 the authors refer to their own work and the work of S. Le Roy. It would be nice if they could provide a reference as it wiould facilitate the understanding of their arguments.
